# Green Chemistry Metrics, A Review

Joel Martínez [1,2], J. Francisco Cortés [3] and René Miranda [1,*]

1 Facultad de Estudios Superiores Cuautitlán, Universidad Nacional Autónoma de México, Cuautitlán Izcalli, Mexico City 54740, Mexico; atlanta126@gmail.com
2 Facultad de Ciencias Químicas, Universidad Autónoma de San Luis Potosí, San Luis Potosí 78210, Mexico
3 Colegio de Ciencias y Humanidades-Azcapotzalco, Universidad Nacional Autónoma de México, Alcaldía Azcapotzalco, Ciudad de México 02420, Mexico; francisco.cortes.ruiz.velasco@gmail.com
* Correspondence: mirruv@yahoo.com.mx; Tel.: +52-55-5623-2056

**Abstract:** Attending both the United Nations Decade of Education for Sustainable Development (2005–2014) and the United Nations 2030 Agenda for Sustainable Development, this review is presented, bearing in mind that green chemistry is essential to contribute to sustainability. This work has compiled all the information relating to green chemistry metrics, so that stakeholders can select an appropriate model, under the Green Chemistry Protocol, to evaluate how much green is a process. The review was organized considering the following convenient sections: the mass valuation, the recognition of the human health and environmental impact, metrics using computational programs (software and spreadsheets), and finally global metrics. This review was developed by consulting the principal databases, since the appearance of the first green chemistry textbook in 1998. A massive number of references were attained involving the keywords proposed below, with six languages observed, highlighted by the English language. It is important to emphasize that the 12 Principles of Green Chemistry are conceptual and offer little quantitative information. In addition, almost all the reported metric green propositions do not consider the 12 principles and few papers offer how to obtain an appropriate evaluation about the greenness of a research. In this sense, it is convenient to note that only in the Spanish literature are there two metrics that consider all the principles. Finally, to our knowledge, and after a deep search in the literature, it is the first review that covers the different features of green chemistry: mass, environment/human health. and in some cases, the use of computational programs.

**Keywords:** review; green chemistry; metrics; sustainable development; 2030 agenda





## 1. Introduction

The Sustainable Development Goals (SDGs), implemented by the United Nations (2015–2030), correspond to a worldwide demand to protect the world, demanding that all people enjoy peace and prosperity [1]. The settled 17 SDGs are interconnected, being essential to note that their achievement in a particular area will engender consequences in others. Complementarily, it is convenient to remember that Sustainable Development (SD) requires the balance of its three pillars (social, economic, and environmental) [2,3].

Interrelated to this work, we, the authors considered mainly two of the seventeen SDGs: Goal 4, referring to the quality of education, ensuring that all learners acquire the knowledge and skills needed to promote SD; and Goal 12, referring to responsible consumption and production reducing waste generation through prevention, reduction, recycling, and reuse [2,3].

On the other hand, green chemistry (GC) has been defined as the use of chemistry for pollution prevention using suitable designs of products and processes, reducing and mainly, if possible, eliminating the use and generation of hazardous substances. The GC has a protocol: the 12 Principles of Green Chemistry, Table 1 [4].

**Table 1.** The 12 Principles of Green Chemistry.

| Principle | Meaning |
|---|---|
| 1 | Prevention: It is better to prevent waste than to treat or clean up waste after it is formed. |
| 2 | Atom economy: Synthetic methods should be designed to maximize the incorporation of all materials used in the process into the final product. |
| 3 | Less hazardous chemical synthesis: Wherever practicable, synthetic methodologies should be designed to use and generate substances that possess little or no toxicity to human health and the environment. |
| 4 | Designing safer chemicals: Chemical products should be designed to preserve efficacy of function while reducing toxicity. |
| 5 | Safer solvents and auxiliaries: The use of auxiliary substances (e.g., solvents, separation agents, etc.) should be made unnecessary wherever possible and innocuous when used. |
| 6 | Design for energy efficiency: Energy requirements should be recognized for their environmental and economic impacts and should be minimized. Synthetic methods should be conducted at ambient temperature and pressure. |
| 7 | Use of renewable feedstocks: A raw material of feedstock should be renewable rather than depleting wherever technically and economically practicable. |
| 8 | Reduce derivatives: Unnecessary derivatization (blocking group, protection/deprotection, temporary modification of physical/chemical processes) should be avoided whenever possible. |
| 9 | Catalysis: Catalytic reagents (as selective as possible) are superior to stoichiometric reagents. |
| 10 | Design for degradation: Chemical products should be designed so that at the end of their function they do not persist in the environment and break down into innocuous degradation products. |
| 11 | Real-time analysis for pollution prevention: Analytical methodologies need to be further developed to allow for real-time, in-process monitoring, and control prior to the formation of hazardous substances. |
| 12 | Inherently safer chemistry for accident prevention: Substances and the form of a substance used in a chemical process should be chosen to minimize the potential for chemical accidents, including releases, explosions, and fires. |

Since its birth (early 1990s), the GC paradigm has reached an imperative status in the chemistry field; in this sense, many educational institutions and industries around the world have implemented the GC Protocol [5] to contribute to SD.

It is appropriate to discuss the main differences between "Green Chemistry" and "Sustainable Chemistry": green chemistry focuses on the design, manufacture, and use of chemicals to decrease pollution potential, according to Anastas and Warner [4]. It "*is an approach that provides a fundamental methodology for changing the intrinsic nature of a chemical product or process so that it is inherently of less risk to human and the environment, to prevent pollution, and thereby solve environmental problems, promoting pollution prevention and industrial ecology*", while sustainable chemistry comprises both the impressions of green chemistry and the effects of processing, materials, energy, and economics [6]. Nevertheless, the meaning of sustainable chemistry is engaged toward the life cycle assessment (LCA), which is associated with the entire life cycle of a product, process, or activity [7,8]. Additionally, in an indirect way, GC is involved in SD because the economic impact is diminished, mainly the regulatory compliance, waste disposal cost, and others. Moreover, the social pillar aspect could attend to reduce the negative image of chemistry employing this strategy due to it being considered as the main contaminant source of the planet.

A metric proposal involves a collection of indicators to provide information on different features of a problem [9]. It must be simple, easily measurable, provide clear information, objective rather than subjective, and undoubtedly defined [10,11]; in this sense, to estimate a process (how green is it?) under the GC Protocol, several metrics have been fashioned. Hence, to establish how green a process is, no one could manage what has not been measured [12].

Our research program is mainly focused on the implementation of green chemical procedures, mostly using non-conventional activating methods of reaction, such as near-infrared and microwave irradiations, ultrasound, and tribochemistry to generate green processes [13–20]. It is important to highlight their corresponding greenness evaluation, by the employment of two metrics proposed by our group [21,22].

Consequently, considering the importance of the application of a convenient green chemical metric, the goal of this work is to produce a review compiling conveniently the corresponding contributions offered in the chemical literature. It is also important to

highlight that according to a deep search and based on our knowledge, this is the first work related to a review connected to holistic greenness evaluations. The compiled information was organized into four convenient sections: firstly, the mass metrics are displayed; secondly, the environment/human health hazard metrics are presented; thirdly metrics using computational programs (software, spreadsheets) are introduced; and finally, the fourth section associates these manuscripts with the global incidence on the 12 Principles of Green Chemistry. It is important to mention that several reviews have been published [23–35]; however, it is essential to highlight that, in general, these works describe the mass metrics or the use of metrics in a specific compound synthesis. A paper by Sheldon in 2018 [36] describes some mass metrics, in addition to sustainability metrics, principally governed by LCA; recently, a review suggested the use of a software approach to determine the sustainability of reactions and processes [37].

The goal of this review is to compile works that include: the mass metrics, the environmental/human health metrics, and, in some cases, the use of computational programs, based on the 12 Principles of GC. This is in addition to suggesting to stakeholders an approach under the Green Chemistry Protocol to evaluate how much green is a chemical process? Additionally, to attend both the United Nations Decade of Education for Sustainable Development (2005–2014) and the United Nations 2030 Agenda for Sustainable Development.

## 2. Methods

The literature search was accomplished by employing the *SciFinder®*, *Scopus*, *Google Scholar*, and *Researchgate* databases. It was performed since the appearance of the first green chemistry textbook, by Anastas and Warner [4] in 1998, considering the following keywords:

Green chemistry approach
Green chemistry evaluation
Green chemistry metrics
Green chemistry measures
How green is?
Which is greener?
Greenness evaluation
Greenness synthesis
Greenness chemistry

## 3. Results and Discussion

Searching in the most important databases, a great number of references were found and then enrolled in appropriated assertions, shown in Table 2. In addition, we wish to welcome the following commentaries (a–d) which comprise general encountered information.

**Table 2.** References in the main search databases [1].

| Keyword | SciFinder® | Scopus | Google Scholar | Researchgate [4] |
|---|---|---|---|---|
| Green chemistry approach | 332 | 716 | 713,000 [2] | 10,200,000 |
| Green chemistry evaluation | 63 | 846 | 1,030,000 [2] | 6,560,000 |
| Green chemistry metrics | 62 | 122 | 394,000 | 680,000 |
| Green chemistry measures | 62 | 24 | 1,220,000 [2] | 14,500,000 |
| How green is? | 49 | 440 | 947,000 [3] | 136,000,000 |
| Which is greener? | 17 | 571 | 166,000 | 154,000,000 |
| Greenness evaluation | 13 | 23 | 23,700 | 250,000 |
| Greenness synthesis | 162,102 | 13 | 17,800 | 121,000 |
| Greenness chemistry | 101,912 | 50 | 17,400 | 158,000 |

[1] Duplicates were not removed. [2] The mean word identified is green chemistry. [3] The mean word identified is green. [4] The largest number of papers are not focused on the use of any metric.

(a)   The obtained information was found in several languages: English, Portuguese, Japanese, Polish, Chinese, and Spanish.

(b)   Linked to the two last keywords, almost all references correspond to examples of application of the metrics.

(c)   The 12 principles are conceptual and do not provide a quantitative framework [38]; however, the encountered works possess both quantitative and/or qualitative features.

(d)   The work is only addressed on the 12 Principles of Green Chemistry, with it important to note that the LCA evaluation was dropped, since to our knowledge it is outside the scope of green chemistry.

### 3.1. Mass Metrics

Next to the disclosure of the Green Chemistry Protocol (the 12 Principles), several metrics were created to evaluate the sort toward green chemistry; in other words, how much green is a process?

In the past 20 years, a considerable number of green chemistry mass metrics have been offered, summarized in Table 3. They aim to highlight those efforts in which two important parameters were mainly considered: atom economy (AE) and the E-factor (E) by Trost [39,40] and Sheldon [41,42], respectively. The AE focused on the maximum number of atoms of reactants appearing in the product [39,40] and the E-factor highpoints the waste minimization and resource efficiency for chemicals, mainly those manufactured in the fine chemicals industry [41,42]. The role of green metrics is more important than ever, highlighting that the mass metrics are focused only on mass.

Some other mass green metrics were encountered after a deep literature search. They are briefly stated and incorporated into Table 3:

i.    The effective mass yield (EMY) was proposed in 1999 [43] for the synthesis of conduritol C and conduritol F, attempting to define the yield considering the percent of mass of desired product with respect to the mass of all-hazard materials used in the synthesis, without considering the mass of benign solvents.

ii.   Curzons et al. [44] reported a green technology guidance involving several metrics: mass intensity, energy pollutants, and toxics. The authors highlighted in the long term, the AE, and in the short term, the solvent use, focusing their use on the reaction and during the workup step to increase the mass intensity value. In addition, solvents are considered for both their development and considerable life cycle impact associated with impacts through use and final disposition.

iii.  In 2002, Ref. [45] defined the entitled metric mass productivity, being the reciprocal of mass intensity (MI), as analogous to effective mass yield and AE. Moreover, it is also defined the AE as including the intermediates in the reaction, in addition to the association with the cost.

iv.   In general, Andraos (2005–2007) combined a set of four metrics [46–48], considering both experimental and calculated parameters and displaying the dependence between them: reaction yield, atom economy, stoichiometric factor (SF), and the corresponding value to the aspect that accounts for the solvent during and post-reaction and/or the catalyst recovery, evaluating linear and convergent sequences. This was in addition to the kinetic resolution of chiral substrates. Andraos [47] involved the construction of a synthesis plan tree to know the efficiency of linear and convergent synthesis by the determination of the gRME. In addition, a study involving costs and quantities in the different steps of a reaction depicted by Andraos and Sayed [48] was performed employing a radial pentagon considering the key metrics; this was to guide bachelor students in understanding the green chemistry concept. Complementarily, Andraos [49], using a pentagram, displayed a simplified approach for a linear and convergent synthesis plan by direct application of green chemistry principles.

v.    Concerning the MI metric, Augé, in 2008 [50], introduced the parameter of the mass of all auxiliaries for linear, convergent, and mixed sequences. It is convenient to

note that the metric is influenced by atom economy, yield, excess of reactants, and mass auxiliaries, indicating that the explanation of environmental (E) impact is produced from auxiliaries.

vi. Quantifying the volume intensity of the solvent consumed (AMVI) and waste generated from the HPLC analytical method, an interesting metric was developed by Hartman et al. [51]. In this study, the authors considered both sample preparation and operation of the analytical device waste, considering the number of samples analyzed, during all the steps of the analysis.

vii. At an industrial level, the American Chemical Society Institute Pharmaceutical Roundtable (the Roundtable) [52,53] implemented the concept process mass intensity (PMI) as the main mass green metric instead of the E-factor, bearing in mind the importance of the process efficiency. In other words, it is preferably an efficient process. Moreover, related to the water, it is convenient to note its inclusion in the PMI metric by the Roundtable, with E excluding it.

viii. A holistic metric for a total synthesis is the global material economy (GME), employed for linear, convergent, and multiconvergent synthesis [54]. This metric is based on the mass of all the required materials to generate the product.

ix. A new metric reaction mass intensity (RMI) was introduced by Song et al. in 2012 [55]; it focuses on the efficiency of the route instead of the process, excluding the solvents. It was defined between total of mass of reaction material and mass of product.

x. Two novel metrics were developed (based on the first two principles of the Green Chemistry Protocol), hybridizing three well-known metrics (AE, MI, and RME), green atomic level, and green mass level [56]. Both cases considered the incorporation of reagents into the product, the theoretical value from AE, and the evaluation of total reaction.

xi. In 2015, Roschangar et al. [12] established the Green Aspiration Level$^{TM}$ (GAL$^{TM}$); it is based on the concept of a modified E, particularly in the pharmaceutical industry. The implementation of the API was launched on the concept of E, considering a complete-E (cE) in later phases. Consequently, knowing the waste stream from solvents and water and the average steps per target drug, it is possible to describe the average chemical transformation (tGAL$^{TM}$) and process (GAL$^{TM}$). Complementarily, to know the green status of a synthetic process, a new metric (relative process greenness (RPG) was developed based on the GAL$^{TM}$. Thus, to determine the RPG, the changes between each phase, using the relative (green) process improvement, RPI, and relative complexity improvement, RCI, are considered.

xii. In 2018 [57], an update of the previously commented metric, iGAL, was made. In this case, the impact of the synthesis plan and the innovations for the process design are highlighted. In this sense, the approach considers three complex parameters (number of fluorine functional groups, rings, and chiral centers) to know the molecular weight (MW) of the drug and its salt-free form (FMW). It is worth noting that the last indicator is considered as the best descriptor related to drug complexity. Consequently, the iGAL metric is employed to determine the RPG parameter to know the greenness of a pharmaceutical process.

xiii. A holistic toolkit (the CHEM 21 project) was proposed (2015) [58] for the pharmaceutical industry; it includes several stages, considering the different parameters covered in each pass. Thus, three new metrics—optimum efficiency (OE), renewable percentage (RP), and waste percentage (WP)—were accomplished. In this sense, the first new metric is dependent on AE which determines the theoretical efficiency, while RME gives the observed value and allows a direct comparison between different reaction types. The second metric is derived from renewable intensity, i.e., the mass of all renewably derivable materials used and the mass of the product. Finally, the third metric is based on the waste intensity and PMI. Using flags with different colors, to determine the hazard of the different steps of the overall process,

a green flag is indicative of the preferred option, an amber flag is an acceptable option but with some issues, and a red flag is assigned to an undesirable process. In this strategy are also considered the solvents, the hazard of reagents, the type of process (batch or flow), and the work steps in the manufacture of the drug. It is convenient to comment that a central objective of designing the toolkit was to allow the recent, state of the art to be assessed for each class of transformation, to know the reaction or pathway, providing a convenient baseline to compare discoveries; in other words, to be an indicator of success. This, in addition, to recognizing hot-spots and bottlenecks in current methodologies helps chemists direct their research to areas of highest effect. A final objective is to inspire constant improvement, and to lead academics to surmise conveniently about sustainability by examining and generating improvements to their synthetic routes.

xiv. A classroom activity that focused on bachelor's students facilitating the teaching–learning of green chemistry was reported in 2016 [59]. In this sense, several reaction metrics were joined (conversion, selectivity, yield, AE, RME, CE, E, and effective mass yield), employing the assembly, alteration, and disassembly of interlocking building blocks for molecular models. In this study, an accomplished model represents a molecule, a brick represents an atom, and the number of connections points represents the molecular weight. The activity is suitable for students in general chemistry courses through advanced undergraduate green chemistry or industrial chemistry courses.

xv. An important study connected to the biopharmaceutical field applying the PMI metric was published in 2019 [60]; its main objective was an application to produce monoclonal antibody (mAb) to quantify the corresponding environmental footprint. The PMI inputs were grouped in four categories: (1) upstream process (cell culture and bioreactor production), (2) harvest (centrifugation and filtration), (3) downstream process (purification train), and (4) drug substance (ultrafiltration/diafiltration and bulk fill), to collect mass data on the amount of water, raw material, and consumables employed to generate 1 kg of API. Accordingly, the authors reported that in large scale, PMI ranged between 3000–24,000, and in small scale, PMI ranged between 3000–17,000. It can be concluded that the efficiency in the production of mAb is not dependent on scale, with it being important to note that the chromatography operation step is the main water consumption, increasing the PMI value.

xvi. As has been previously asserted, the PMI green metric is considered the key metric for an industrial pharmaceutical process. Thus, in recent research (2020), two synthetic processes were compared to determine their green qualifications [10]. The authors pondered three parameters—yield, concentration, and the difference in molecular weight of the reactants—as being important to highlight that the corresponding difference must be as close as practically possible, instead of considering only a mass metric such as AE or E, to obtain a more realistic response.

xvii. A novel metric manufacturing mass intensity (MMI) was recently developed [61] to measure the impact of producing API through a synthetic process. The metric is an extension of the PMI metric. It considers the categories excluded in PMI, such as cleaning/preparation, equipment conditioning, effluent management, abatement, overages, circularity, reuse, and recycling. With the MMI metric, it is possible to know the mass requirements to prepare the equipment, verify if some reagent or solvent might be replaced, and know if it is possible to reuse (recycle) any solvent or reagent, and recognize the accurate mass quantification to avoid more optimizations in the process.

**Table 3.** Mass metrics to evaluate green chemistry.

| Mass Metrics | Expression | References |
|---|---|---|
| E-factor (E) | Total waste/products | [12,41,42,52,53,59,60] |
| Atom economy (AE) | (FW product/FW of all reactants used in reaction) × 100 | [12,39,40,44,45,49,56,59,60] |
| Mass intensity (index) (MI) | Total mass/mass of product | [12,44,50,52–54,56,60] |
| Mass productivity | Mass of product/total mass in process or process step | [45] |
| Effective mass yield (EMY) | (Mass of products/mass of non-benign reagents) × 100 | [12,43,44,59] |
| Generalized reaction mass efficiency (gRME) | Mass of product/total mass used in a process or process step | [46–49] |
| Reaction mass efficiency (RME) | (Mass of isolated product/total mass of reactants used in reaction) × 100 | [12,44,45,54,56,59,60] |
| Stoichiometric factor (SF) [1] | | [46–49] |
| Carbon efficiency (CE) | (Mass of carbon in product/total mass of carbon in reactants) × 100 | [12,44,45,59,60] |
| Process mass intensity (PMI) | Total mass in a process or process step/mass of product | [12,49,52,53] |
| Global material economy (GME) | Mass of product/total mass used in total synthesis | [54] |
| Reaction mass intensity (RMI) | Total of mass of reaction materials/mass of product | [55] |
| Greener atomic level | 100 × (RME/AE) | [56] |
| Greener mass level [2] | | [56] |
| Complete E-factor (cE) | $\sum m$(Raw materials) + $\sum m$(reagents) + $\sum m$(solvent) + $\sum m$(water) − $m$(product)/$m$(product) | [12] |
| Transformation green aspiration level[TM] (GAL[TM]) | $x$E/average complexity, where $x$E = cE o E | [12] |
| Green aspiration level[TM] (tGAL[TM]) | (tGAL[TM]) × complexity | [12] |
| Relative process greenness (RPG) | GAL($x$F)/$x$E | [12] |
| Innovation green aspiration level (iGAL) | iGAL = (mGAL × FMW)/1000 | [57] |
| Optimum efficiency (OE) | (RME/AE) × 100 | [58] |
| Renewable percentage (RP) | (Renewable intensity/PMI) × 100 | [58] |
| Manufacturing mass intensity (MMI) | U(PMI) + U(MI of component under consideration) [3] | [61] |
| Scale risk index (SRI) | SRI = $t$ (HF$_R$ + HF$_P$ + HF$_D$) | [62] |

[1] Considering reactions run under nonstoichiometric conditions. [2] not reported. [3] U = utilization factor of component.

### 3.2. Environmental-Human Health Metrics

i. A set of indicators assembled in a hierarchical metric has been proposed to evaluate the greenness of new and existing chemical technologies, as well as for the analysis of flow sheets of new processes. In this sense, two criteria were developed: (a) it should enable comparison of alternative technologies, and (b) ideally, it should provide a single framework, applicable across the chemical industries with an appropriate selection of indicators, considering four vertical hierarchy levels: 1. product and process, 2. company, 3. infrastructure, and 4. society. It is important to note that the selection of indicators depends on each industrial sector and which their products consider the most relevant aspects oriented to green chemistry in vertical hierarchy levels: 1. climate change, 2. continuous availability of energy, 3. footprint resource efficiency, 4. product toxicity, 5. minimum use of solvents, and 6. recyclability, among others [9].

ii. In 2008 [63], a study to produce nanomaterials was reported; this work coupled both industrial engineering and green chemistry, determining the environmental impact for the synthesis of silica nanoparticles, implicating: a sol-gel process, a simple flame- and an improved HMDSO-flame method. In general, the practice involves two components, shown in Figure 1. The first component involves the development of a set of sustainability metrics and, in a second place, taking up the corresponding decision applying the metrics to evaluate the rate of the process. It is convenient to

consider that the sustainability metrics for industrial engineering are process and safety parameters, such as heat of the reaction, chemical interactions, toxicity, pressure, and temperature, among others. For green chemistry, environmental impact metrics include solvent index, generation of waste, and AE, among others.

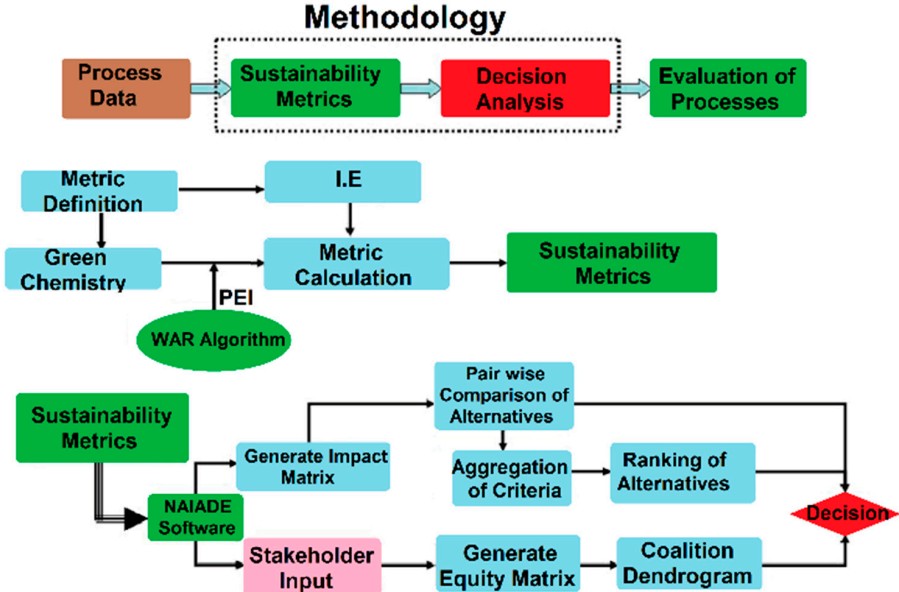

**Figure 1.** Principal components of the methodology. Adapted from Ref. [63].

iii. During the performance of a study related to the polymers field, Ref. [64] evaluated the efficacy of both the 12 green engineering, Table S1, and the green chemistry principles with LCA. In this sense a set of 12 polymers was studied considering the metrics AE, mass from renewable sources, biodegradability, recycled percent, the distance of furthest feedstock, price, life cycle health hazard, and life cycle energy use. Thus, it was reported that the 12 polymers from petroleum or biological categories represent a good option toward the green design principles, reducing the environmental impact.

iv. Environmental matters are a world concern; in this sense, Mercer et al. reported in 2012 [65] an instance created for undergraduate and graduate students to determine the greenest of several syntheses. In this case, nine metrics from LCA were employed: acidification potential, ozone depletion potential, smog formation potential, global warming potential, human toxicity by ingestion, human toxicity by inhalation, persistence, bioaccumulation, and abiotic resource depletion potential, in addition to AE and E. This study reported that the LCA metrics are more reliable than AE and E metrics. Consequently, these calculations allow students to make environmental decisions to attach a label of green to a particular chemical process.

v. In 2012, Gałuszka et al. [66] proposed a semi-quantitative Eco-Scale as a novel approach to evaluate the greenness of an analytical methodology, based on assigning penalty points to parameters of an analytical process without agreement with the ideal green analysis. This green analytical methodology involves the following facts: 1. sample collection, 2. preservation, 3. transport and storage, 4. sample preparation, 5. calibration, and 6. validation of analytical methods. The basis of the concept of an analytical Eco-Scale is that the ideal green analysis has a value of 100, but in a more realistic case, >75 represents excellent green analysis, and <50 represents inadequate green analysis. This simple analytical Eco-Scale is a good semi-quantitative tool for laboratory practice and educational purposes.

vi. A new benign index (BI) [67] based on the following environmental hazards—acidification–basification, ozone depletion, global warming, smog formation, inhalation toxicity, ingestion toxicity, inhalation carcinogenicity, ingestion carcinogenicity, bio-concentration, abiotic resource depletion, cancer potency, persistence, and endocrine disruption—

was announced by Andraos, vide supra. It is important to mention that this index is defined as a fraction between 0 and 1, and it may be added as another radial axis to previous work [33] to evaluate the green merits of any given chemical reaction. Importantly, this index is applicable to waste, input, and output materials.

vii. In the same way, the safety/hazard index (SHI) was introduced [68], considering the following environmental impacts: corrosive gas, corrosive liquid/solid, flammability, oxygen balance applied to combustion reactions and oxidation reactions, hydrogen gas generation, occupational exposure limit, and dermal absorption, among others. In addition, there is the temperature and pressure hazard. These indexes are applicable to single- and multiple-step synthetic plans.

viii. An interesting hybrid comparison was developed at the University of Toronto [69] between Eco-Scale, Green Star method, and BI and SHI indexes, considering the following parameters: reaction temperature, reaction pressure, $LD_{50}$ (oral and dermal), flammability, and corrosivity, among others. Employing a red–yellow–green–gray color code, where the score is achieved by the mass contribution of waste substance and the number of color cells accrued for each substance, this score is also applicable for the input materials.

ix. Another interesting study relates to microscale (at a laboratory level) [62], though in this analysis the microscale did not improve safety. Thus, a new scale risk index was introduced (SRI), shown in Table 3. Its purpose is to assess the improvement of safety on reducing the scale of synthesis experiments. This index includes the following variables: 1. hazard of substances, 2. the time of exposure to substances, and 3. the amounts of substances used. It is important to note that this index is a direct metric of the risk, being an inverse metric of safety and benignity.

x. In 2018, Ref. [70] published a paper in the analytical area that used the green analytical procedure index (GAPI) to evaluate the green character of an entire analytical methodology, from sample collection to final determination, focusing on the 12 green analytical principles, shown in Table S2. In this methodology, five pentagrams were proposed to evaluate and quantify the environmental impact for each step of any analytical methodology, shown in Figure 2. The GAPI tool is a pictogram determining the greenness of each stage in the analytical procedure, employing both a color scale and three levels of evaluation for each stage. The color scale is from green through yellow to red to quantify low, medium, and high environmental impact, respectively, for each step, remarking that this tool is more efficient comparing different procedures. Additionally, the circle in middle GAPI is related to a procedure for qualification and quantification; thus, the GAPI does not show the circle where a procedure is only for qualification.

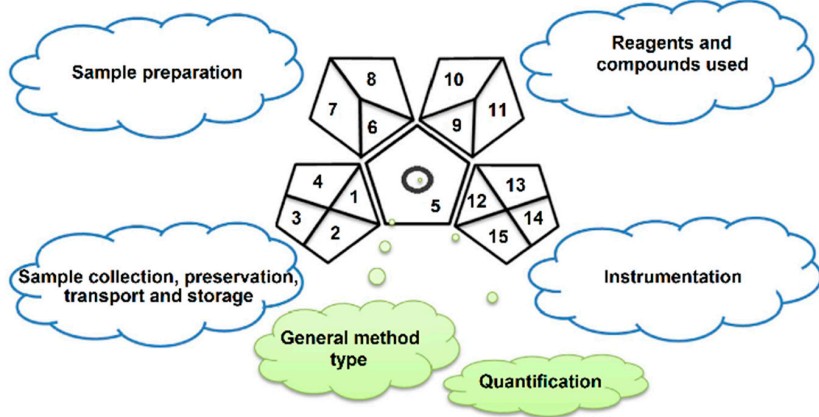

**Figure 2.** GAPI pictogram. The numbers in pentagrams are related to parameters description. Adapted from Ref. [70].

It must be mentioned that the above metrics avoid the evaluation of the mass (AE, E, MI, and RME, among others), and, in general, several metrics of this section could be considered as part of LCA.

### 3.3. Metrics Using Computational Programs (Software, Spreadsheets)

To facilitate the understanding of green chemistry metrics and as a complement to mass or environmental metrics, many works have employed spreadsheet templates and software strategies.

i.    In 2007, Andraos and Sayed [48] employed a spreadsheet template, shown in Figure 3, to calculate the RME for chemical transformations at the laboratory level. In this sense, the parameters immersed in RME calculation (AE, reaction yield ($\epsilon$), stoichiometric factor (SF), and material recovery parameter (MRP)) are displayed in a radial pentagon. Each axis corresponds to one of five parameters arising from the center with values ranging between 0 and 1, shown in Figure 4. Therefore, the best green condition is caused by a regular pentagon and the worst green situation is represented by the distorted pentagon [71]. Complementarily, an upgrade study, employing a radial hexagon was introduced by Andraos in 2012 [67].

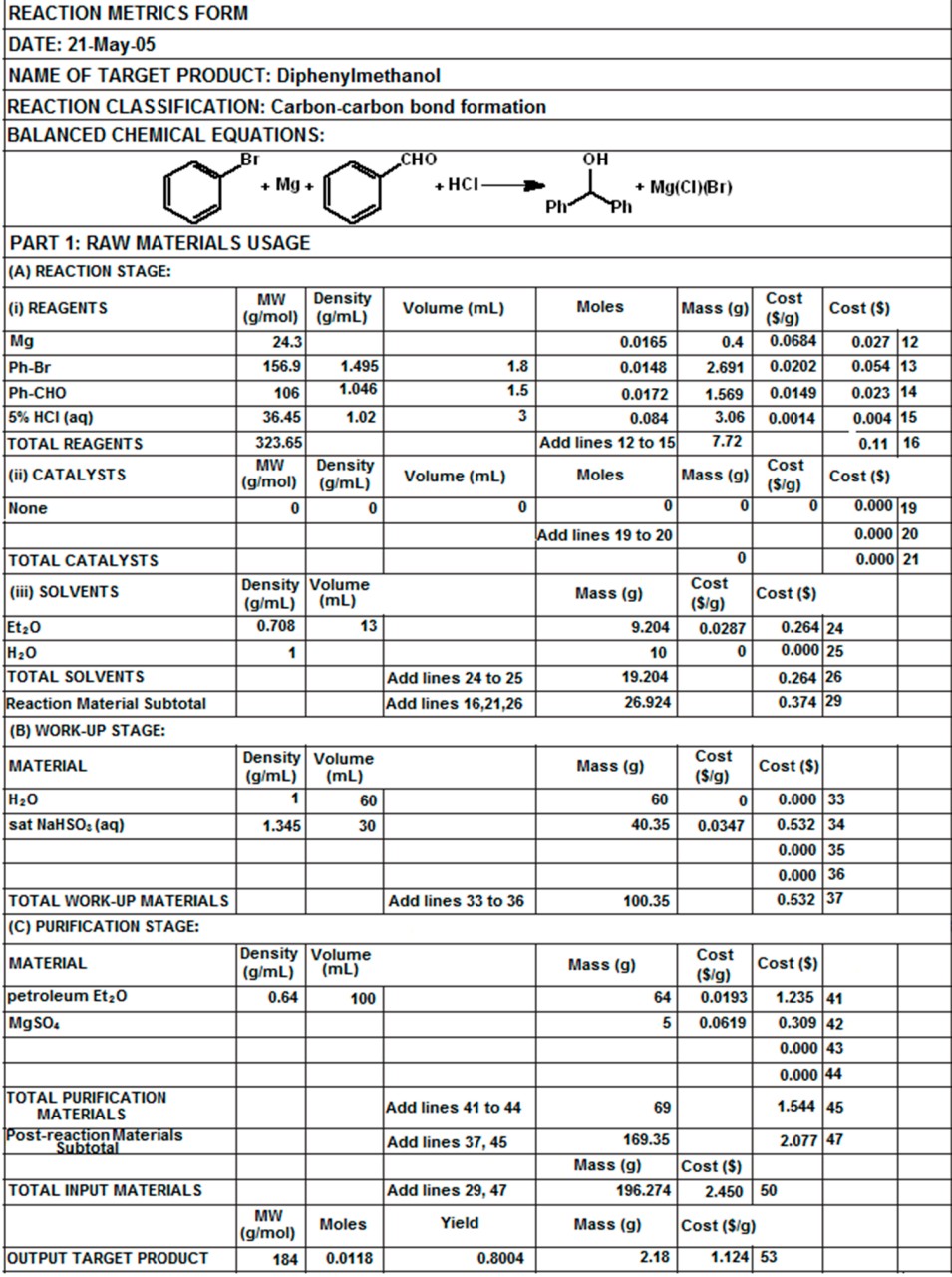

**Figure 3.** *Cont.*

| PART 2: GREEN METRICS ANALYSIS | | | | | | | | | |
|---|---|---|---|---|---|---|---|---|---|
| **Limiting reagent:** | Ph-CHO | | | | | | | | |
| **PARAMETER** | VALUE | | | | | | | | |
| Reaction scale | 0.0148 | moles | | 58 | | | | | |
| E(mw) | 0.759 | MW byproducts/ | | 59 | | | | | |
| | | MW product | | | | | | | |
| AE | 0.569 | MW product/ | | 60 | | | | | |
| | | SMW reagents | | | | | | | |
| (i) Under reclaiming reaction solvents, catalysts, and byproducts, and all post-reaction materials | | | | | | | | | |
| Mass of waste | 5.54 | g | | 63 | | | | | |
| (line 16-53) | | | | | | | | | |
| E(m) | 2.541 | g waste/ g product | | 64 | | | | | |
| RME | 0.282 | g product/S g reagents | | 65 | | | | | |
| SF | 1.611 | | | 66 | | | | | |
| Waste input cost ($) | 0.079 | | | | | | | | |
| (ii) Under committing all reaction solvents, catalysts, and byproducts, and post-reaction materials to waste | | | | | | | | | |
| Mass of waste | 194.094 | g | | 69 | | | | | |
| (line 50-53) | | | | | | | | | |
| E(m) | 89.034 | g waste/ g product | | 70 | | | | | |
| RME | 0.011 | g product/S g reagents | | 71 | | | | | |
| Wasted input costs ($) | 2.423 | | | | | | | | |
| Check formula | 0.011 | | | | | | | | |
| (iii) Under reclaiming ether from reaction and petroleum ether from purification | | | | | | | | | |
| Mass of waste | 120.89 | g | | 76 | | | | | |
| E(m) | 55.454 | g waste/ g product | | 77 | | | | | |
| RME | 0.018 | g product/S g reagents | | 78 | | | | | |
| Wasted input costs ($) | 0.934 | | | | | | | | |

**Figure 3.** Reaction metrics to calculate the RME. Adapted from Ref. [48].

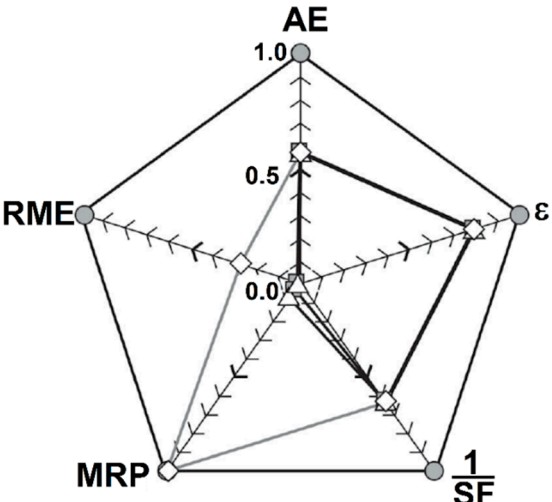

**Figure 4.** RME metric values represented by radial pentagon. Adapted from Refs. [48,71].

ii.   A strategy, coupling both industrial engineering and green chemistry, was performed to determine the environmental impact during the synthesis of silica nanoparticles by sol-gel or flame processes, using as complements the Waste Reduction Algorithm (WAR) and Novel Approach to Imprecise Assessment and Decision Environments (NAIADE) software [63], shown in Figure 5. It is important to note that the employed green chemistry metrics (material procurement, generation of waste, hazardous material, AE, solvent index, and energy efficiency) were employed to determine the environmental impact of three nanoparticles synthesis (HMDSO, flame-TEOS, and sol-gel methods) using the WAR and NAIADE software strategies.

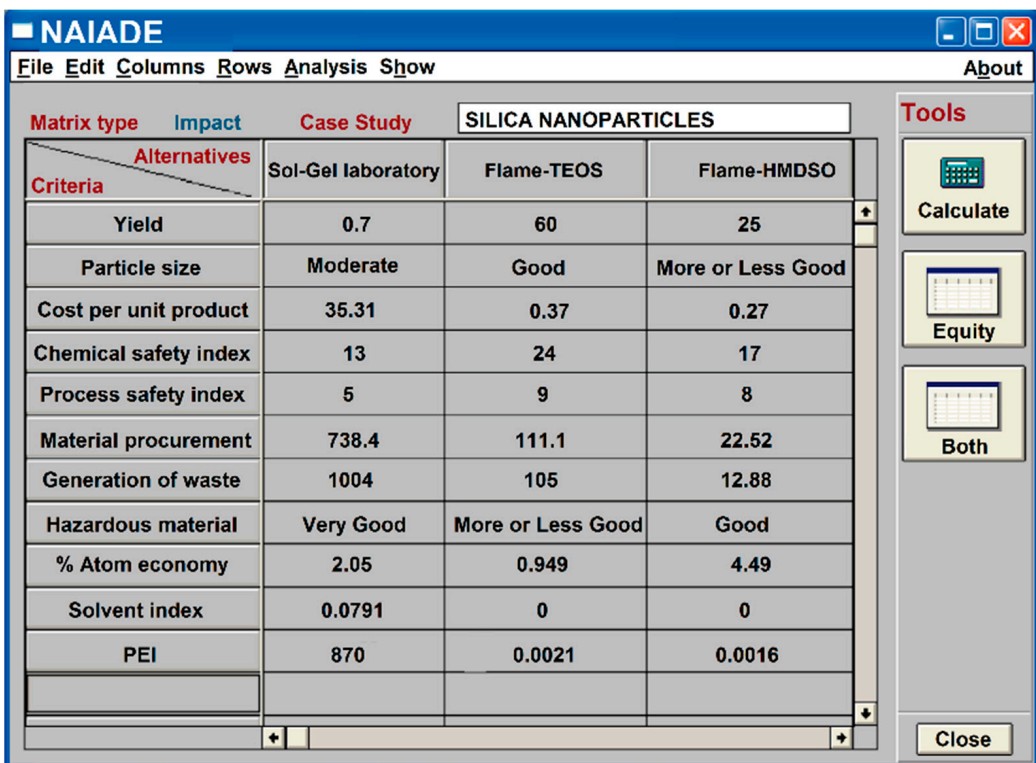

**Figure 5.** Matrix generated from the NAIADE software. Adapted from Ref. [63].

iii. A semi-quantitative metric to recognize the greenness of a reaction, entitled green star (GC), was developed by Ribeiro in 2010 [72]. The 12 principles are involved, and an Excel radar chart is employed, shown in Figure 6. The scale proposed to establish the greenness of a reaction comprises values from 1 to 3. This approach evaluated the substances involved in a process, considering both the risk to human health and the environment.

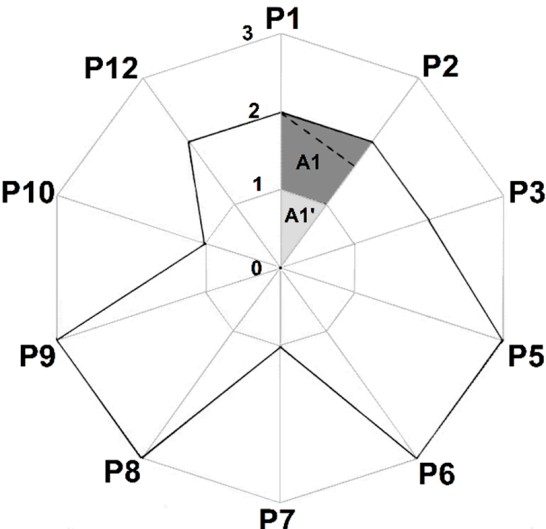

**Figure 6.** GS model using the 12 principles (P1-12). A1 and A1' meaning the area of triangle. Adapted from Ref. [70].

iv. The PMI implemented by the Roundtable [52,53] as the main mass green metric, at the Pharmaceutical–industrial level since 2011 has a novel version known as the PMI-spreadsheet calculator tool, shown in Figure 7. It was developed by Merck to recognize the process mass intensity across the entire pharmaceutical supply chain.

This procedure involves the substrates, reagents, and solvents, among other stages in the API's development [52]. It must be mentioned that this tool has been accomplished with LCA to know the environmental impact, but it is essential to remark that the LCA data are unavailable for many solvents; consequently, some features of nearest solvent are necessary.

| Step Name/Number | 1 | |
|---|---|---|
| | Value | Units |
| Physical Batch Size | 155 | kg |
| Assay Purity | 99% | wt% |
| Assay Batch Size | 153.5 | kg |
| Yield | 91% | |
| Assay Kg product | 217 | kg |
| Product Purity | 100% | wt% |
| | | |
| Raw Materials | Physical Charge (kg) | Units |
| Substrates | | |
| Product from step 1 | 155 | kg |
| Reagents | | |
| Diisopropylethylamine | 105 | kg |
| 4-chlorobenzoyl chloride | 147 | kg |
| | | |
| Solvents | | |
| 2-MeTHF | 700 | kg |
| Heptane | 450 | kg |
| | | |
| Aqueous | | |
| 2N HCl | 420 | kg |
| 25% NaCl | 220 | kg |
| | | |
| | | |
| **PROCESS STEP METRICS** | | |
| Mass Substrate (kg) | | 155 |
| Mass Reagents (kg) | | 252 |
| Mass Solvents (kg) | | 1150 |
| Mass Aqueous (kg) | | 640 |
| Step PMI | | 10.1 |
| Step PMI Substrate, Reagents, Solvents | | 7.2 |
| Step PMI Substrate and Reagents | | 1.9 |
| Step PMI Solvents | | 5.3 |
| Step PMI Water | | 2.9 |
| Cumulative PMI | | 19.0 |
| Cumulative PMI Substrate, Reagents, Solvents | | 14.0 |
| Cumulative PMI Substrate and Reagents | | 3.2 |
| Cumulative PMI Solvents | | 10.8 |
| Cumulative PMI Water | | 5.0 |

**Figure 7.** The Roundtable PMI-spreadsheet calculator tool. Adapted from Ref. [53].

v. To measure the health, safety, and environmental impacts for the flavor and fragrance industry, the GREEN MOTION$^{TM}$ metric tool was established [73]. It is a simple and quantitative method; however, it displays several limitations, such as the lack of the analysis of manufacturing processes and that the penalty points (0–100) were arbitrarily selected by the authors. The core of this metric are seven concepts: raw material, solvent, hazard and toxicity of the reagents, reaction, process, hazard and toxicity of the final products, and waste. It was created by a simple answer (yes or no) from a total of 100 points; for each question answered with negative impacts, penalty points are subtracted.

vi. Recently, both the ChemPager tool [74] and an improved version [75] were launched, based on Google sheets or an Excel spreadsheet, and Tibco Spotfire platform for the required visualization. The convenient use of this tool offers appropriate summarized information of the valued project; consequently, the assistant chemist could conveniently review the process to make a suitable decision. In addition, the ChemPager displays important data of a PMI metric, solvent distribution, production cost, volume-time yield, and a performance score, shown in Figure 8. The data allows distinguishment of the variability of the process, if any component is changed, for example, the equivalents of catalyst in a batch process.

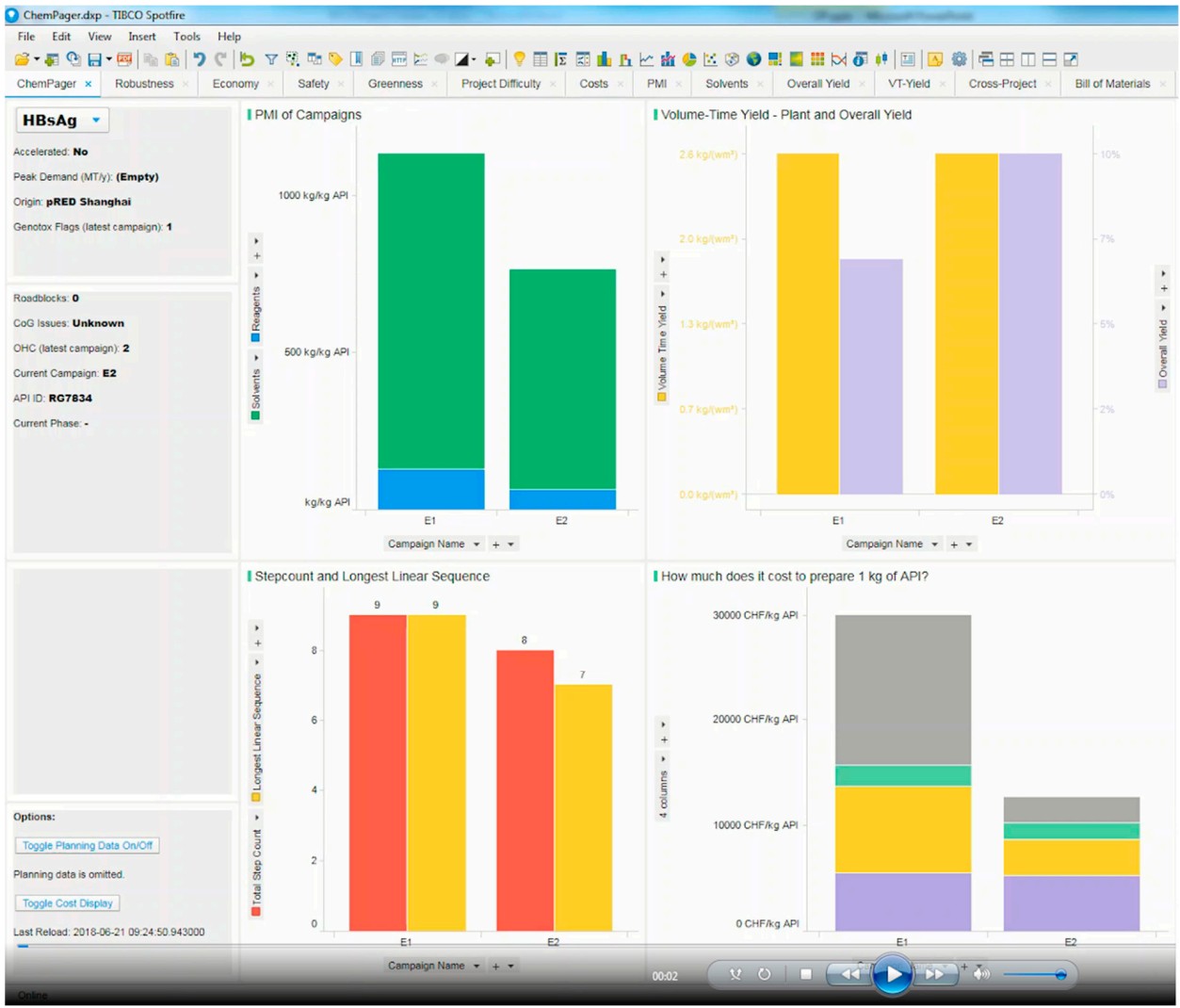

**Figure 8.** *Cont.*

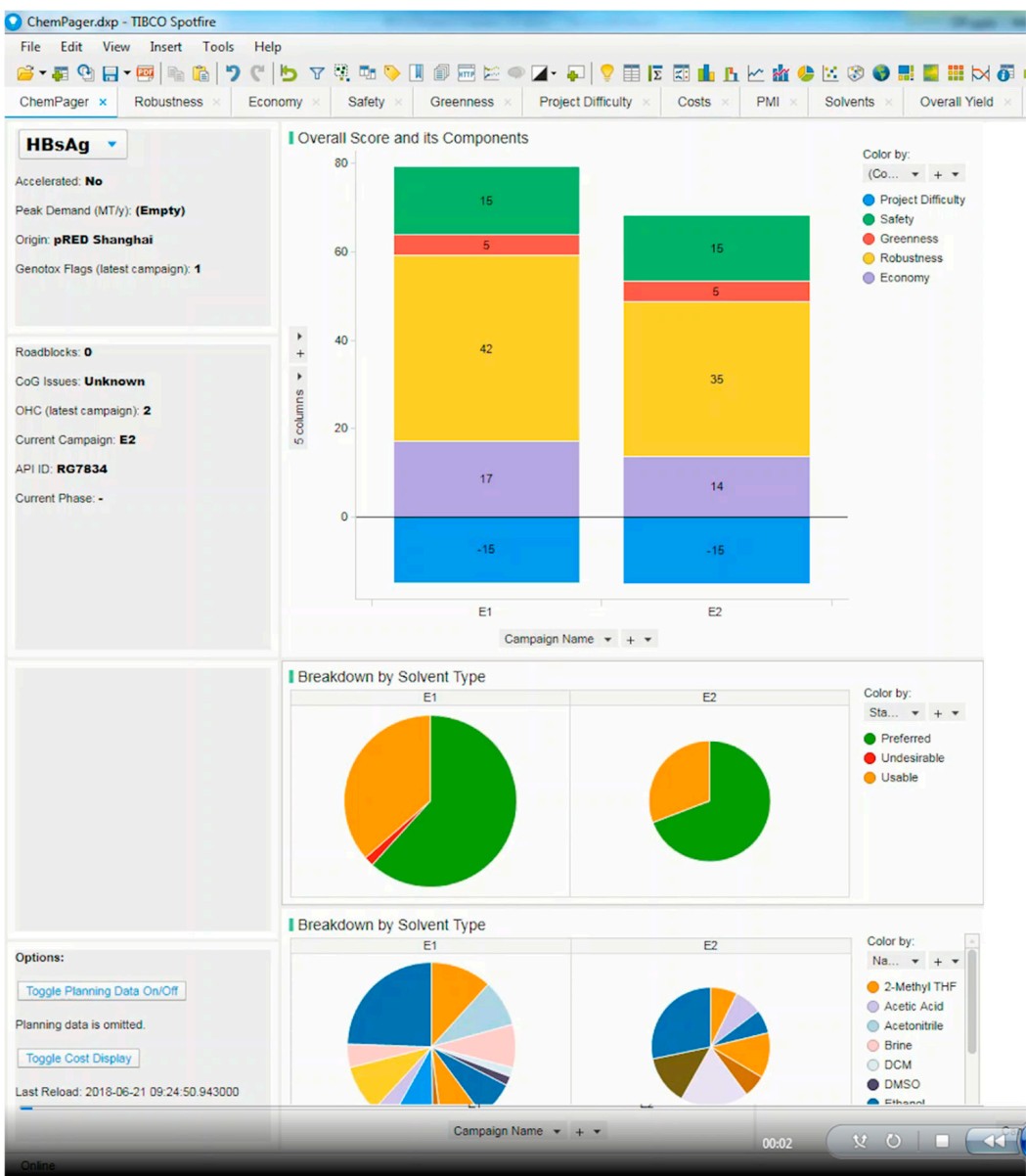

**Figure 8.** Example of page in ChemPager, partial view. Adapted from Refs. [74,75].

vii. The use of "big data" to determine the hazard and risk of the chemicals has been important in silico tools to know their toxicology. In this sense, several data sources (ECHA, ToxCast, HSDB, and ACToR) display, for more than 1000 chemicals, their toxicological features to elucidate the molecular mechanism of toxicity. Moreover, some computational approaches have been applied: Ecosar calculates the physical properties and the potential ecotoxicity values, OncoLogic predicts the cancer hazard, the Scivera approach uses curated data to derive scores for individual endpoints, and the Verisk 3E GreenScore methodology displays hazard scores, using logarithmic transformation of raw scientific data [76].

viii. Related to the field of analytical chemistry, a color metric scale has been reported to evaluate any analytical procedure [77], employing an Excel spreadsheet. This tool is based on the three primary colors (red, green, and blue) to define a color scale, proportioning a final evaluating color. In this case, the red color is attributed to the analytical performance, the green color for safety/eco-friendliness, and the blue color for productivity or practical effectiveness. In general, the quantitative range corresponds to 0–100%. In this sense, if the color scale is ≥66.6%, it indicates

a satisfying value; if the obtained value is between 33.3% and 66.6%, it is labeled as tolerance range; finally, if the color scale is <33.3%, it indicates a lack of acceptance. In addition, it is important to mention that this metric is assisted using quantitative parameter W (method brilliance) which integrate the three-color scales with different weights and the parameter w (criteria) that can be adjusted according to the situation.

ix. An interesting green metric tool, entitled AGREE (analytical greenness) software [78], appeared in recent research. It focuses on the 12 principles of green analytical chemistry (GAC), shown in Table S2. In the AGREE metric, a scale of 0–1 is employed, and the result is offered by a pictogram, shown in Figure 9, where the scale values are designated by the user. It is important to note that all 12 principles of GAC have been assigned a scale, from sample treatment (direct input is preferred), the size of the sample, the use of energy (minimized), to the safety of the operator.

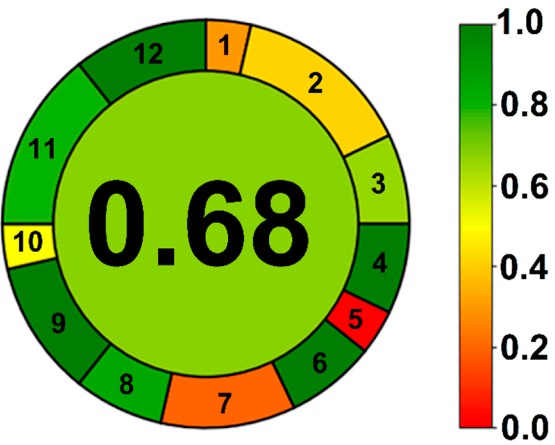

**Figure 9.** Pictogram showing the result of assessment and color scale. Adapted from Ref. [78].

x. A convenient update of the above-commented work [70] was recently published (2021) by Płotka-Wasylka and Wojnowski [79]; it was named the Complex Green Analytical Procedure Index (ComplexGAPI). In this work, a pentagram–pictogram, a color scale, and two or three levels of evaluation are employed. Additionally, a hexagon–pictogram was implemented, shown in Figure 10, complementing the evaluation: yield, reagents, solvents, conditions, instrumentation, workup, and purification of the final product.

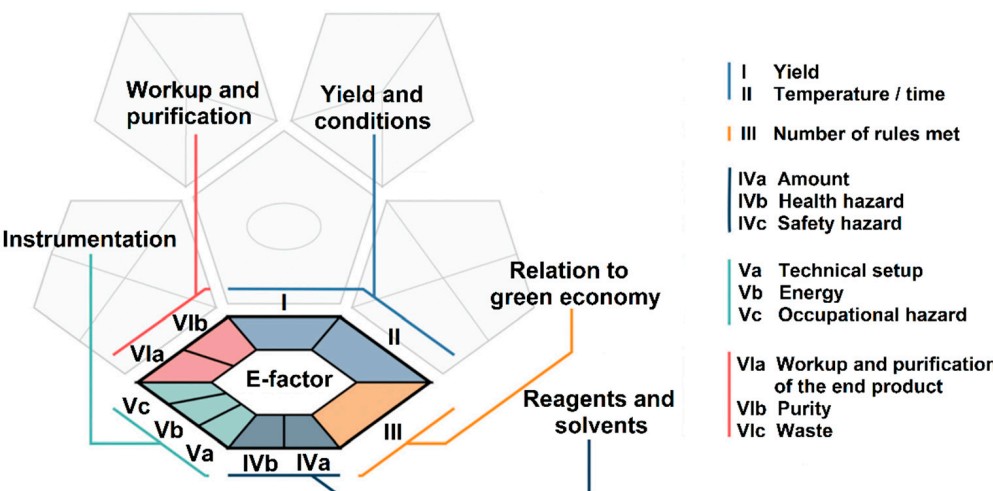

**Figure 10.** The ComplexGAPI pictogram. Adapted from Ref. [79].

xi.　An interesting study is the application of a multicriteria decision analysis (MCDA) to determine the greenest organic synthesis procedures, employing the TOPSIS algorithm (Technique for Order of Preference by Similarity to Ideal Solution) [80]. This proposal was established with nine criteria: 1. reagent, 2. reaction efficiency, 3. atom economy, 4. temperature, 5. pressure, 6. synthesis time, 7. solvent, 8. catalyst, and 9. reactant. This metric is based on a scale from 0 to 10 points, with the criteria assigned by two experts according to their perception, aided by the material safety data sheets of reagents and reactants and a Globally Harmonized System of classification and labeling of chemicals (GHS), being a more systemic tool because it combines criteria into a final score.

xii.　Also related to the GAC area, it is important to highlight that the sample preparation is a main step in the separation of the analyte. In this sense, there was a report of the first work with a metric tool (AGREEprep) [81] that gives importance to sample preparation. It was fashioned considering 10 categories, based on the principles of the green sample preparation, shown in Table S3 [82]. It is also employed a scale subscore from 0 to 1, with a further final qualitative calculation score. The result is visualized by mean of a colorful pictogram with the final evaluation value, localized inside a colored circle in the center of the pictogram, displaying the overall sample preparation greenness performance, Figure 11.

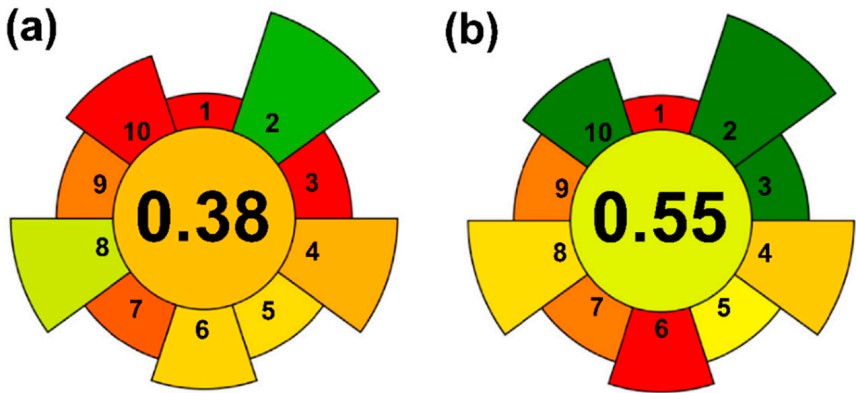

**Figure 11.** Example of AGREE prep results for phthalate esters preparation by: (**a**) dispersive liquid–liquid microextraction, (**b**) solid phase microextraction. Adapted from Ref. [81].

### *3.4. Global Metrics*

As can be perceived according to all the information previously described, no one of the presented metrics judged the whole Green Chemistry Protocol (the 12 principles); therefore, in this section are considered those metrics that comprise the target protocol.

i.　A metric labelled as Eco-Scale, planned by Van Aken et al. [83], uses a scale from 0 to 100, with 100 the value for an ideal reaction and, consequently, 0 indicates a failed reply. The assigned value is based on outcomes that consider six parameters: yield, price of reaction components, safety, technical configuration, temperature/time, treatment and purification, shown in Table 4. It is important to note that each parameter comprises individual cumulative penalty points. The corresponding analyses are straightforward given that all important parameters are transparent (it is clear how the final score is obtained), are fast, can be calculated in less than 5 min, and do not take a general standpoint but consider advantages and disadvantages of specific methodologies or auxiliary reagents, in addition to offering a general overview of the reaction conditions.

**Table 4.** The penalty points to calculate the Eco-Scale. Data from Ref. [83].

| Parameter | Penalty Points |
|---|---|
| Yield | |
| Price of reaction components (to obtain 100 mmol of end product) | |
| Inexpensive (<$10) | 0 |
| Expensive (>$10 and <$50) | 3 |
| Very expensive (>$50) | 5 |
| Safety | |
| N (dangerous for environment) | 5 |
| T (toxic) | 5 |
| F (highly flammable) | 5 |
| E (explosive) | 10 |
| F + (extremely flammable) | 10 |
| T + (extremely toxic) | 10 |
| Technical setup | |
| Common setup | 0 |
| Instruments for controlled addition of chemicals | 1 |
| Unconventional activation technique | 2 |
| Pressure equipment, >1 atm | 3 |
| Any additional special glassware | 1 |
| (Inert) gas atmosphere | 1 |
| Glove box | 3 |
| Temperature/time | |
| Room temperature, <1 h | 0 |
| Room temperature, <24 h | 1 |
| Heating, <1 h | 2 |
| Heating, >1 h | 3 |
| Cooling to 0 °C | 4 |
| Cooling, <0 °C | 5 |
| Workup and purification | |
| None | 0 |
| Cooling to room temperature | 0 |
| Adding solvent | 0 |
| Simple filtration | 0 |
| Removal of solvent with bp < 150 °C | 0 |
| Crystallization and filtration | 1 |
| Removal of solvent with bp > 150 °C | 2 |
| Solid phase extraction | 2 |
| Distillation | 3 |
| Sublimation | 3 |
| Liquid–liquid extraction | 3 |
| Classical chromatography | 10 |

ii.   In 2010, an interesting semi-quantitative metric entitled Green Star (GS) was re-
ported [72,84]. According to the authors, this graphical metric helps decide the most
acceptable reaction, considering the 12 principles. The graphical metric consists of
a star, with each corner associated with one of the 12 principles, shown in Figure 6,
linking a punctuation from 1 to 3, according to the risks for both human health and the
environment. This is in addition to pondering chemical accidents, and degradability
and renewability of the substances involved; moreover, combining mass metrics (yield
%, MI, AE, EF, and RME). Consequently, this metric offers a visual analysis, allowing
the optimization of the process by inspection, to improve the greenness of the process.
However, it is important to highlight that the fourth and eleventh principles are not
considered. In the same context, two other semi-quantitative metrics were settled:
the green circle (GrC) and the green matrix [85,86]. In this case, the GrC, shown in
Figure 12, is divided in 12 sections. Each section is colored according to its agreement
with a particular principle, allowing the identification of those features that can be
improved which, in the case of the green matrix, are considered the strengths and
weaknesses, and the advantages and disadvantages, of a chemical process.

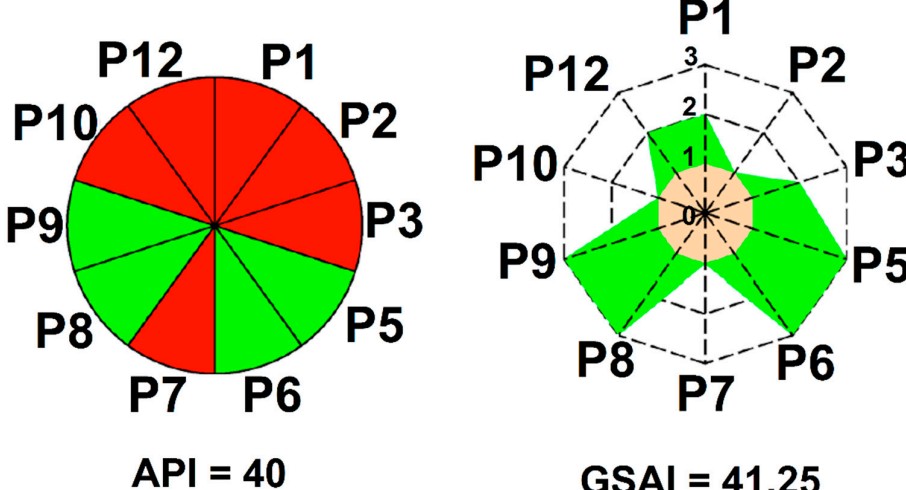

**Figure 12.** Example of green metric in comparison with green star metric. The green color is for the accomplished principle, the red color, if it is not. Adapted from Refs. [70,84–87].

iii.      An update of the previously discussed metrics was reported in 2014 by Ribeiro et al. [87]. In this case, the use of both GS and GrC in conjunction with GHS was offered, displaying advantages such as a more systemic assessment of the hazards of chemicals for use in evaluation criteria and increased contact with GHS in the laboratory context.

iv.      In 2011, an interesting hybrid tool to evaluate experimental procedures related to green chemistry was reported [21]. In this concern, a green metric pondering all 12 principles of the GC was presented. This approach is qualitative by means of a color code and semi-quantitative by using a Likert-type scale: from totally brown (number 1) to completely green (number 10), as shown in Figure 13. As can be seen, clear assignment (increasing vs. diminishing, the green contribution) is offered by considering several green color tones, a yellow color as interface, and the various brown color tones. To complement the green-degree evaluation of a particular process, a flowchart of the complete experimental procedure is displayed, step by step (reaction, isolation, and purification of a product), as illustrated in Figure 14. It is important to note that health and safety pictograms of reagents and solvents are also considered during the evaluation. Each principle is considered in every step (step by step), with the corresponding value score placed in parentheses linking the involved principle by its number in a box color according to the green intensity assigned to the corresponding experimental step. The concluding greenness value is the average result of all the steps judged in the entire experimental procedure.

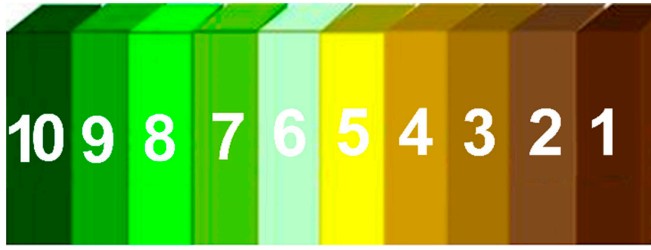

**Figure 13.** Color-Likert scale for evaluation of green chemistry principles. The numbers meaning: 1 = totally brown, 2 = very brown, 3 = moderately brown, 4 = slightly brown, 5 = brown to green transition, 6 = slight green approach, 7 = good green approach, 8 = very good approach, 9 = great green approach, 10 = totally green. Adapted from Ref. [21].

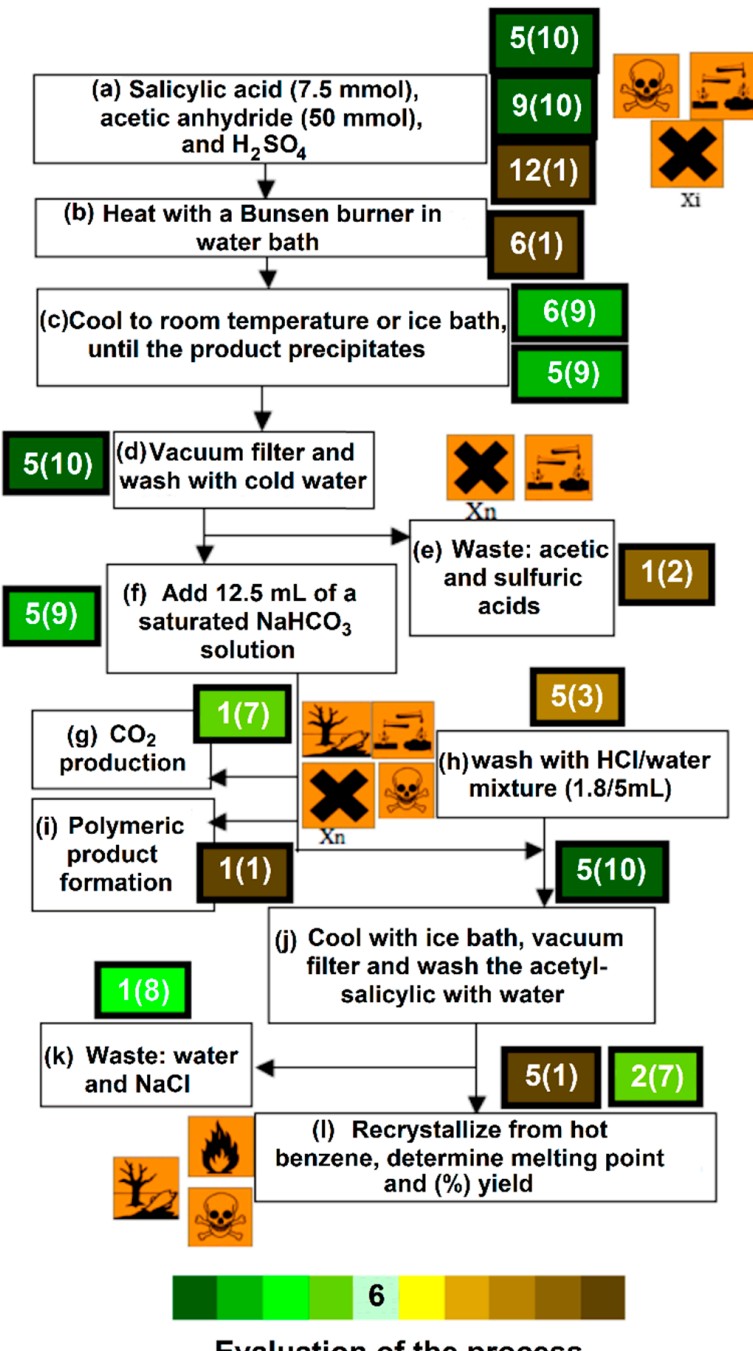

**Figure 14.** Example of flowchart of the experimental procedure using the Color-Likert scale for evaluation of green chemistry principles. Adapted from Ref. [21].

Also, it is appropriate to point out that in this green metric, the Toxics Release Inventory (TRI) list is also considered [88].

v.    In 2013, Ref. [89] was launched the iSUSTAIN® tool developed by the chemical industry. It involved the 12 Principles of GC to measure the sustainability of products and processes. The 12 principles are updated and measured using a score range between 0 and 100, with 100 corresponding to the best score. Some principles such as 3, 5, 6, 8, 9, 11, and 12 are evaluated according to the original literature. However, principle 1 is measured by EF, principle 2 by reaction mass efficiency, principle 4 is a mixed metric (aquatic toxicity and human toxicity), principle 7 is the sum of the weight of renewable raw materials in a product per weight of the product, and

principle 10 is measured by actual experimental data. This approach aids researchers to improve a product, since they can detect the stage with the lowest score in the process and consequently make further modifications, contributing to generating better sustainability for a particular product.

vi.    Another important contribution was reported by Duarte et al. [90]. In this work, the authors analyze the greenness of a synthesis describing the total process in a separate assessment of the different steps—reaction, isolation, and purification—as well as the global process, shown in Figure 15. The methodology involves the 12 Principles of Green Chemistry and GS [72,84–87] methodology, as described above. In this case, regarding the GS, the metric uses a ten-corner star for the reaction and six corner star for the workup operations. It is convenient to highlight that the authors made this assumption: "*It is possible to improve the greenness of a synthesis without performing laboratory work, just by identifying the best performing reaction, isolation, and purification steps.*"

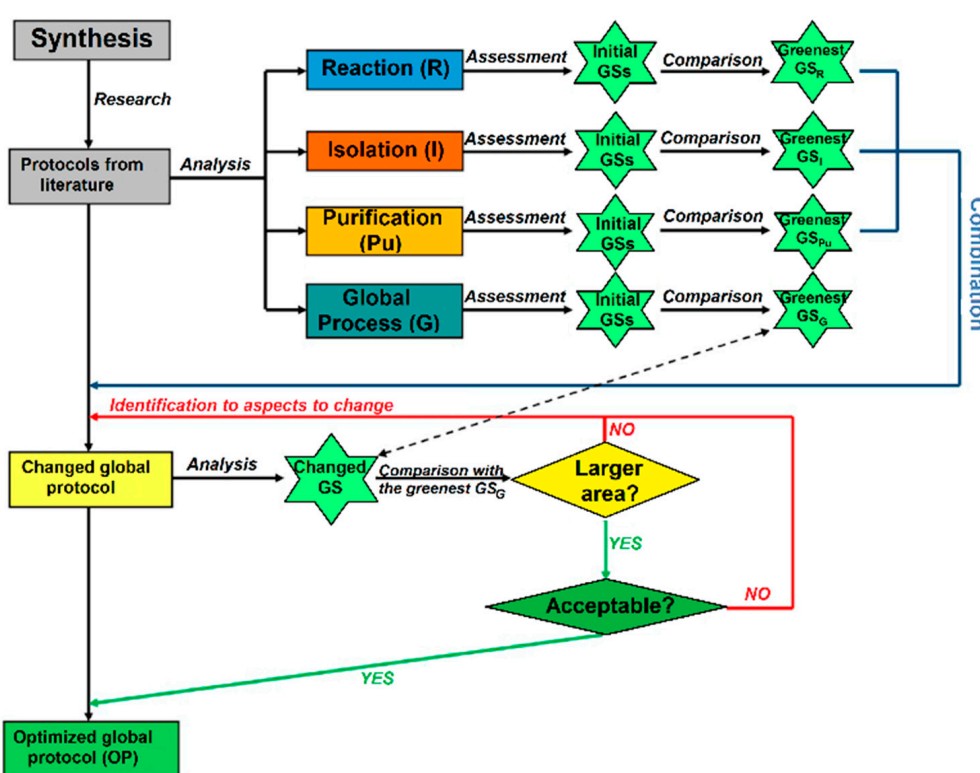

**Figure 15.** GS procedure considering all the steps in the synthesis. Adapted from Ref. [90].

vii.    An interesting work employing a traffic-light tactic to assert a green metric methodology, in other words the greenness of a process, was reported in 2016 [22]. By this practice, 13 color tones were engaged, wandering between green–yellow–red, as in traffic-light devices, shown in Figure 16. It is important to highlight that the 12 Principles of Green Chemistry are considered. In addition, to allow a suitable mode for the greenness of a particular process, an experimental flowchart is depicted.

In the corresponding diagram, the treatment and disposal of waste—in addition, the risk to human health and the environment are considered—as a complement, the pictograms of GHS and fire diamond of NFPA are appropriately applied. In conclusion, depending on the resulting color tone, a greenness degree of the experiment is afforded. It is convenient to note that the resulting color tones are obtained using both RGB (red, green, black) and CMYK (cyan, magenta, yellow, black, key) models from computational evaluation codes.

| N.° GCPAE | Color tone | RGB color model | CMYK color model |
|---|---|---|---|
| 0 | | 237, 28, 37 | 0.00, 0.882, 0.844, 0.071 |
| 1 | | 240, 81, 35 | 0.00, 0.663, 0.854, 0.059 |
| 2 | | 237, 110, 5 | 0.00, 0.54, 0.98, 0.07 |
| 3 | | 243, 146, 0 | 0.00, 0.40, 1.00, 0.05 |
| 4 | | 249, 179, 0 | 0.00, 0.28, 1.00, 0.02 |
| 5 | | 255, 210, 0 | 0.00, 0.18, 1.00, 0.00 |
| 6 | | 255, 237, 0 | 0.00, 0.07, 1.00, 0.00 |
| 7 | | 239, 227, 0 | 0.00, 0.05, 1.00, 0.06 |
| 8 | | 214, 217, 0 | 0.01, 0.00, 1.00, 0.15 |
| 9 | | 187, 207, 0 | 0.10, 0.00, 1.00, 0.19 |
| 10 | | 157, 196, 26 | 0.20, 0.00, 0.87, 0.23 |
| 11 | | 122, 185, 41 | 0.34, 0.00, 0.78, 0.27 |
| 12 | | 79, 174, 50 | 0.55, 0.00, 0.71, 0.32 |

N.° GCPAE: Green chemistry principle according to the experimentation

**Figure 16.** Traffic light metric to assess the green approach. Adapted from Ref. [22].

## 4. Future Recommendations

The main objective of this review is to exhibit the published modes to evaluate how green a practice is, in addition to the fact that about 80% of published works indicate that they are "green". However, no explanations are provided; we are convinced that novel metrics with holistic character are welcome. Moreover, this paper offers an invitation to attend both the United Nations Decade of Education for Sustainable Development (2005–2014) and the United Nations 2030 Agenda for Sustainable Development.

## 5. Conclusions

Given that the role of green chemistry and consequently the requirements of greenness metrics are currently more important than ever, this work, following a profound search in the literature, in addition to an appropriated analysis of the corresponding papers, is to our knowledge the first review related to a greenness metric being offered. Consulting the principal databases from 1998 (the appearance of the 12 principles) to date, a great number of references were obtained, involving the keywords proposed. An interesting fact was that *SciFinder*® and *Scopus* databases offered a lesser quantity of references; however, they better fitted the established keywords. Additionally, to our knowledge, more than 80% of published works indicate that they are "green", but no explanation is given.

An important and welcome conclusion is that green chemistry is an excellent approach that contributes to sustainability. Hence, it is worth highlighting that both the United Nations Decade of Education for Sustainable Development (2005–2014) and the United Nations 2030 Agenda for Sustainable Development were attended.

**Supplementary Materials:** The following supporting information can be downloaded at: https://www.mdpi.com/article/10.3390/pr10071274/s1, Table S1: The 12 green analytical chemistry principles; Table S2: The 12 green analytical principles; Table S3: The 10 green principles for the sample preparation. References [78,81,82,91] are cited in the supplementary materials.

**Author Contributions:** Conceptualization, R.M.; searching, J.M.; methodology, J.M., J.F.C. and R.M.; validation, J.M., J.F.C. and R.M.; formal analysis, J.M., J.F.C. and R.M.; investigation, J.M., J.F.C. and R.M.; data curation, J.M., J.F.C. and R.M.; writing—original draft preparation, J.M. and J.F.C.; writing—review and editing, J.M., J.F.C. and R.M.; visualization, R.M. and J.M.; supervision, R.M.; project administration, J.M., J.F.C. and R.M.; funding acquisition, J.F.C. and R.M. All authors have read and agreed to the published version of the manuscript.

**Funding:** This research was funded by INFOCAB-UNAM, grant number PB200121. R. Miranda also acknowledges Cátedras de Investigación-FESC-2022CI2218.

**Data Availability Statement:** The data reported in this study are available upon request to atlanta126@gmail.com or mirruv@yahoo.com.mx.

**Conflicts of Interest:** The authors declare no conflict of interest.

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
