# Peer review of "Green Chemistry Metrics, A Review"

_processes, doi:10.3390/pr10071274_

Round 1

Reviewer 1 Report

--

Reviewer 2 Report

The article is a review on metrics applied in green chemistry. It is another review on this topic but it has some merits and can be potentially interesting for the readers. However, before publication some points have to be improved or clarified.

In the introduction it would be good to state the relation between green chemistry and sustainable development. GC relates only to environmental aspects of SD.

L 41 should be “Since its birth”

Table 1 lists only titles of GC principles. It should give also short explanations on principles meanings.

Section 3 – what is an implication of this bibliometric analysis? It is presented without importance for further contents.

L171 what are “greenery” reagents? Change to “green”

L 283 analytical eco-scale is presented in this section, while organic synthesis eco-scale is presented in another one. Their mechanism is virtually the same.

Figure 7 Fill this figure with data to make it case study. It will be more clear and beneficial for better reader understanding

3.5. section – this section is not needed. It is a summary without drawing conclusions.

The reference to recent Green Chem  https://doi.org/10.1039/D1GC03108B article is missing. It should be discussed as another approach.

As there are other reviews on this topic, the need to publish this one should be stated.

Reviewer 3 Report

Manuscript ID: processes-1756576

Title: Green chemistry metrics, a review

Journal: Processes

Comments to authors:

General: In the present study, all the information is compiled related to the metrics employed to evaluate how much green is a process, considering the following sections: the mass valuation, the recognition of the human health and environmental impact, metrics using computational programs (software, spreadsheets), global metrics and finally a comparative citation index of global metrics.

Recommendation: This manuscript can be accepted after a minor revision... The authors are requested to address the following comments:

ü  The Abstract should be enriched with the brief details of the methodology.

ü  The authors should also present some numeric results in the Abstract.

ü  English proofreading should be done for grammar and typos such as “Thus, in this work, is compiled all the information related to the metrics employed to evaluate how much green is a process, considering the following sections: the mass valuation, the recognition of the human health and environmental impact, metrics using computational programs (software, spreadsheets), global metrics and finally a comparative citation index of global metrics.”

ü  The novelty, scope, and significance of the present work should be highlighted in the last paragraph of the Introduction section.

ü  The authors are recommended to add more literature review on numerical modeling of relevant works.

ü  What is the need for this work?

ü  The literature review should be improved by adding latest references and discussion.

ü  Results and discussion section must be improved by adding more details and more discussion.

ü  Fig. 4 and 6 can be presented in an improved way.

ü  Results section should be defended using technical reasons and relevant references. More discussion should be added.

ü  Conclusions look like a lab report. They should be refined and briefly presented. Some numerical results should be added.

ü  The authors should add the future recommendations based on the present study.

Round 2

Reviewer 2 Report

The authors have responded to all querries and the quality of manuscript is improved enough to be published in the present form.